# GSK3β Inhibition Ameliorates Atherosclerotic Calcification

**DOI:** 10.3390/ijms241411638

**Published:** 2023-07-19

**Authors:** Xinjiang Cai, Yan Zhao, Yang Yang, Xiuju Wu, Li Zhang, Jocelyn A. Ma, Jaden Ji, Kristina I. Boström, Yucheng Yao

**Affiliations:** 1Division of Cardiology, David Geffen School of Medicine at UCLA, Los Angeles, CA 90095-1679, USA; 2The Molecular Biology Institute at UCLA, Los Angeles, CA 90095-1570, USA

**Keywords:** vascular calcification, glycogen synthase kinase-3β, atherosclerosis

## Abstract

Endothelial-mesenchymal transition (EndMT) drives endothelium to contribute to atherosclerotic calcification. In a previous study, we showed that glycogen synthase kinase-3β (GSK3β) inhibition induced β-catenin and reduced mothers against DPP homolog 1 (SMAD1) in order to redirect osteoblast-like cells towards endothelial lineage, thereby reducing vascular calcification in *Matrix Gla Protein (Mgp)* deficiency and diabetic *Ins2^Akita/wt^* mice. Here, we report that GSK3β inhibition or endothelial-specific deletion of GSK3β reduces atherosclerotic calcification. We also find that alterations in β-catenin and SMAD1 induced by GSK3β inhibition in the aortas of *Apoe^−/−^* mice are similar to *Mgp^−/−^* mice. Together, our results suggest that GSK3β inhibition reduces vascular calcification in atherosclerotic lesions through a similar mechanism to that in *Mgp^−/−^* mice.

## 1. Introduction

Vascular calcification is a common feature of atherosclerotic lesions [1,2]. Recent studies have shown that vascular calcification is an active process involving ectopic bone formation, where osteogenic differentiation occurs in cells transitioning from other lineages [3]. The vascular endothelium is a major contributor of osteoprogenitors to vascular calcification through endothelial-mesenchymal transitions (EndMTs) [4,5]. In this process, endothelial cells (ECs) lose their cell identity but gain plasticity and differentiate into osteoblast-like cells. This dramatic switch of cell fates has been shown to cause calcification in atherosclerotic plaques [6,7]. However, it is unknown if reversing this switch by inducing osteoblastic-endothelial transition ameliorates atherosclerotic calcification. Prior investigations have shown that small molecules are able to reprogram and modulate cell fates [3] and that endothelial-like cells can be derived from other lineages [8]. Recently, we screened more than 20,000 small molecules and found that SB216763, an inhibitor of glycogen synthase kinase 3 (GSK3), reduced EC-derived osteogenic differentiation and decreased aortic calcification in the matrix Gla protein null (*Mgp^−/−^*) mouse, an established model of vascular calcification [9]. Previous studies have demonstrated similarities between the mechanisms underlying vascular calcification in the *Mgp^−/−^* mice and atherosclerotic lesions [6,7]. In this study, we hypothesize that GSK3 inhibition ameliorates calcification of atherosclerotic lesions.

## 2. Results

### 2.1. GSK3β Inhibition Reduced the Calcification of Atherosclerotic Lesions

To test our hypothesis, we used *Apoe*^−/−^ mice, a well-established model of atherosclerosis, that were fed a western diet for 8 weeks. We subsequently used SB216763 (5 µg/g daily) to treat the *Apoe^−/−^* mice for 8 additional weeks with continued western diet feeding. We examined the atherosclerotic lesions and found a significant reduction in lesion calcification (Figure 1a,b). We also examined the aortic expression of the osteogenic markers Osteopontin, Osterix, Osteocalcin, and Cbfa1. Real-time PCR showed that SB216763 dramatically decreased the aortic expression of these markers in the western diet-fed *Apoe^−/−^* mice (Figure 1c). Real-time PCR also showed that SB216763 abolished the aortic induction of the mesenchymal markers Slug, Sca1, c-kit, and Klf4 (Figure 1d). Following a similar design, we treated *Apoe^−/−^* mice with different doses of SB216763 (1–5 µg, daily). We found that SB216763 dose-dependently reduced aortic calcium in the Apoe^−/−^ mice that had been fed a western diet (Figure 1e). The reduction of aortic Osteopontin also correlated with the increase in SB216763 dose (Figure 1f). The results suggested that GSK3 inhibition reduced EndMTs and atherosclerotic calcification.

### 2.2. Endothelial Specifc GSK3β Deletion Limited the Calcification of Atherosclerotic Lesions

GSK3α and GSK3β are the two isoforms of GSK3, and SB216763 inhibits these isoforms in an ATP-competitive manner [10]. We previously used VE-cadherin^CreERT2^ mice in a lineage-tracing approach that identified osteoblast-like cells in aortic tissues [11] and showed that limiting GSK3β shifted osteoblast-like cells towards the endothelial lineage, thereby reducing vascular calcification in *Mgp^−/−^* mice [9]. To determine the effect of limiting GSK3β on atherosclerotic calcification, we generated *VE-cadherin^CreERT2^ GSK3β^flox/flox^Apoe^−/−^* mice. At 10 weeks of age, we injected the mice with tamoxifen for 5 days. Ten days after injection, we fed the mice a western diet for 16 weeks. Von Kossa staining showed reduced calcification in the atherosclerotic lesions of the tamoxifen-treated group (Figure 2a). The reduction in total aortic calcium deposition confirmed the decrease in aortic calcification in the *VE-cadherin^CreERT2^ GSK3β^flox/flox^Apoe^−/−^* mice after deletion of GSK3β (Figure 2b). Real-time PCR and immunoblotting showed that the deletion of GSK3β decreased the osteogenic and mesenchymal markers in the aortic tissues (Figure 2c and Figure 3). We showed that limiting GSK3β increased β-catenin that directly targeted SMAD Family Member 1 (Smad1) to shift osteoblast-like cells towards the endothelial lineage [9]. Here, we examined β-catenin and Smad1 in the aortic tissues of *VE-cadherin^CreERT2^ GSK3β^flox/flox^Apoe^−/−^* mice. Immunoblotting showed an increase in β-catenin together with decreased Smad1 in the tamoxifen-treated group (Figure 3). Together, our results suggest that GSK3β inhibition redirects osteogenesis to ameliorate vascular calcification in atherosclerotic lesions.

## 3. Discussion

Previous studies have shown that EndMTs occur in atherosclerotic lesions in fat-fed *Apoe^−/−^* mice and suggested that increased BMP activity is the primary force driving EndMTs leading to calcification in atherosclerotic lesions [6]. In addition, inhibition of serine proteases or reduction of Sox2 expression, both of which reduced EndMTs, limited the calcification in *Apoe^−/−^* mice [6]. Further studies also revealed that the aortic osteoblast-like cells were redirected back to the endothelial lineage by the SB216763 treatment in *Mgp^−/−^* mice and diabetic *Ins2^Akita/+^* mice [4,9,11]. A similar mechanism underlies the vascular calcification in *Mgp^−/−^* mice, diabetic *Ins2^Akita/+^* mice, and atherosclerotic lesions in *Apoe^−/−^* mice [6,7]. In this study, we show that GSK3β inhibition or endothelial-specific deletion of GSK3β ameliorates atherosclerotic calcification. The results not only support that the same mechanism promotes calcification in these three mouse models but also show the importance of GSK3β inhibition in atherosclerotic calcification and provide new information for developing treatment strategies.

EndMTs drive the endothelium to contribute cells for osteogenic differentiation in the calcifying aorta. It has been shown that mesenchymal markers are expressed in the progression of EndMTs, such as Klf4, c-kit, Slug, Twist, N-cadherin, Sca1, CD90, CD44, CD10, and CD71 [12]. It would be interesting to determine these markers after GSK3β inhibition in calcified atherosclerotic lesions and explore the possibility that the osteoblastic-endothelial transition also passes through a mesenchymal stage. 

GSK3 is a serine/threonine kinase that is constitutively activated in cells [13]. Previous studies have demonstrated that osteogenic and endothelial differentiation are affected differently by GSK3 activity, which promotes osteogenic differentiation [14] but prevents endothelial differentiation [15,16]. GSK3 deficiency disrupts the differentiation of osteoblasts [17], and inhibition of GSK3 promotes endothelial differentiation, proliferation, and migration. SB216763 is a small molecule that specifically inhibits the activity of the two isoforms of GSK3, GSK3a, and GSK3β [10,13]. Here, we show that specific inhibition of GSK3β is responsible for the reduction in lesion calcification. 

SMAD1 is part of the SMAD family of transcriptional factors, which has eight family members, SMAD1-8 [18]. SMAD1 is activated by TGFβ/BMP signals through phosphorylation, and activated SMAD1 is essential for osteoblastic differentiation [19]. SMAD1 activity directly affects osteoblastic differentiation by regulating the differentiation of osteoprogenitor cells [20,21]. In turn, the levels of SMAD1 protein are regulated by GSK3 activity [22]. β-catenin is a member of the catenin protein family that is expressed in many tissues. β-catenin is a mediator of the canonical Wnt signal pathway, which is essential for endothelial differentiation [23]. GSK3 activity directly regulates the level of β-catenin [23,24]. Previous studies have suggested that GSK3β inhibition decreases SMAD1 but increases β-catenin leading to osteoblastic-endothelial transitions that reduce vascular calcification in *Mgp^−/−^* mice and diabetic *Ins2^Akita/+^* mice [4,9,11]. Here, we show similar alterations in the levels of SMAD1 and β-catenin in *Apoe^−/−^* mice after limiting GSK3β, suggesting again that coupled changes in SMAD1 and β-catenin cause the shift of osteoblast-like cells and reduce the calcification.

Other lineages of vascular cells are also known to contribute to vascular calcification, such as smooth muscle cells, pericytes, and fibroblasts [1]. Although this study suggests a transition of osteoblastic fate towards endothelial differentiation, it would be interesting to examine whether GSK3 inhibition causes lineage transitions in other vascular lineages involved in vascular calcification. 

## 4. Materials and Methods 

### 4.1. Animals

*Apoe^−/−^* (B6.129P2-Apoetm1Unc/J) and *GSK3β^flox/flox^* (B6.129 (Cg)-Gsk3b^tm2Jrw^/J) mice were obtained from the Jackson Laboratory. The *VE-cadherin^cre/ERT2^* mouse was obtained as a gift from Dr. Ralf Adams. Genotypes were confirmed by PCR [25], and experiments were performed with generations F4–F6. Littermates were used as wild-type controls. At 8 weeks of age, all *Apoe^−/−^* mice were switched to a high-fat/high-cholesterol diet (western diet) (Research Diets, New Brunswick, NJ, USA, diet #D12108, containing 21% fat, 1.25% cholesterol). Since similar atherosclerotic calcification occurred in both male and female *Apoe^−/−^* mice [7], we used mixed genders in each experimental group. We used the pwr R package to calculate the effect size and statistical power, as we previously published [7]. The number of animals in each group was sufficient to reach more than 80% power and identify the difference in the formation of atherosclerotic calcification. The studies were reviewed and approved by the Institutional Review Board and conducted in accordance with the animal care guidelines set by the University of California, Los Angeles. The investigation conformed to the National Research Council, *Guide for the Care and Use of Laboratory Animals, Eighth Edition* (Washington, DC, USA: The National Academies Press, 2011). SB216763 (Sigma-Aldrich, S3442. Saint Louis, MO, USA) was injected via a tail vein or retro-orbital injection (5 µg/g, daily) as in previous studies [26]. The injections in the *Apoe^−/−^* mice started at 16 weeks of age and continued for 8 weeks. Tamoxifen (Sigma-Aldrich, T5648) was injected for 5 days at 75 mg/kg daily.

### 4.2. RNA Analysis

Real-time PCR analysis was performed as previously described [9]. Glyceraldehyde 3-phosphate dehydrogenase (GAPDH) was used as a control gene. Primers and probes for mouse osterix (Mm00504574_m1) and osteopontin (Mm00172574_cn) were obtained from Applied Biosystems (Waltham, MA, USA) as part of Taqman^®^ Gene Expression Assays.

### 4.3. Immunoblotting

Immunoblotting was performed as previously described [15]. Equal amounts of cellular protein or tissue lysates were used. Antibodies to SMAD1 and GSK3β (Cell Signaling Technology, Danvers, MA, USA, 9743, 93115), β-catenin (R&D System, Minneapolis, MN, USA, AF1329), and osterix (Santa Cruz Biotechnology, Dallas, TX, USA, sc-22536) were used. β-Actin (Sigma-Aldrich, A2228) was used as a loading control.

### 4.4. Quantification of Aortic Calcium

Total aortic calcium was measured using a calcium assay kit (Bioassay, Hayward, CA, USA) as previously described [9]. The frozen tissues were lyophilized and weighed. The calcium was released from the tissues by incubating the samples in 0.6 N HCl at 37 °C for 24 h. After incubation, the tissues were cut into as small pieces as possible, vortexed, and centrifuged. The supernatants were collected in new, pre-labeled tubes. The samples were stored at 4 °C until the calcium assay was carried out.

### 4.5. Lesion Quantification

The mice were euthanized and then fixed by perfusion with 10% buffered formalin via the left ventricle for 4 min. The proximal aortas were excised. The specimens were embedded in OCT (Tissue-Tek, Fisher Scientific, Hampton, NH, USA), frozen on dry ice, and stored at −80 °C until sectioning. Serial cryo-sections were prepared, and every fifth 10 μm section was collected on poly-D-lysine-coated slides. The sections were stained with hematoxylin, Oil Red O, or von Kossa staining. The slides were examined by light microscopy, and the lesion areas were quantified with computer-assisted image analysis (Image-Pro Plus, Media Cybernetics, Bethesda, MD, USA).

### 4.6. Oil Red O Staining

Frozen sections were air dried and fixed in formalin, then briefly washed with running tap water for 1–10 min. The sections were subsequently rinsed with 60% isopropanol and stained with a freshly prepared Oil Red O working solution for 15 min. The sections were then rinsed with 60% isopropanol and stained lightly for nuclei with alum hematoxylin (5 dips). Finally, the sections were rinsed with distilled water and mounted in aqueous mounting or glycerin jelly.

### 4.7. Von Kossa Staining

Once the sections had been deparaffinized and hydrated to water, they were incubated with a 1% silver nitrate solution and placed under sunlight for 1–2 h. The sections were rinsed with distilled water, then washed with 5% sodium thiosulfate for 5 min to remove un-reacted silver. The sections were further rinsed in distilled water and counterstained with Nuclear Fast Red for 5 min. Finally, the sections were dehydrated through graded alcohol, cleared in xylene, and covered with permanent mounting medium.

### 4.8. Statistical Analysis

Data were analyzed for statistical significance by ANOVA with post-hoc Tukey’s analysis or Student’s *t*-test. The analyses were performed using GraphPad Instat^®^, version 9.0 (GraphPad Software). The data represent mean ± SD. *p* < 0.05 was considered significant, and experiments were performed a minimum of three times.

## Figures and Tables

**Figure 1 ijms-24-11638-f001:**
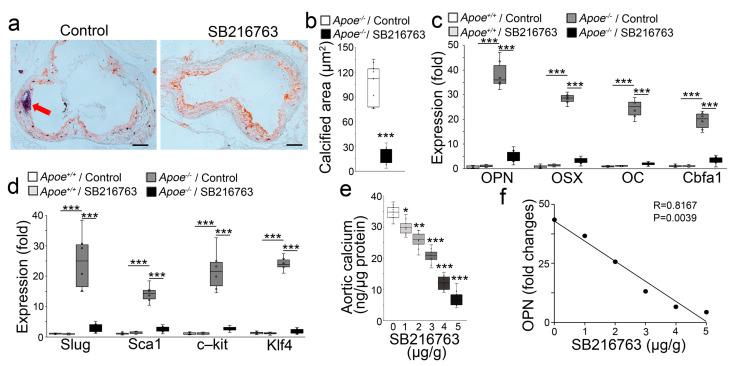
GSK3 inhibition ameliorates vascular calcification in atherosclerosis. *(***a**) Oil Red O staining of aortic sinus sections of the proximal ascending aortas of *Apoe^−/−^* mice fed a western diet, with or without SB216763 treatment. Saline was used as a treatment control. Scale bar, 100 µm; red arrows indicate calcification. (**b**) Total aortic calcium of the proximal descending aortas of *Apoe^−/−^* mice fed a western diet, with or without SB216763 treatment (n = 6). Saline was used as treatment control. (**c**,**d**) Expression of osteogenic markers Osteopontin (OPN), Osterix (OSX), Osteocalcin (OC), Cbfa1, and mesenchymal markers Slug, Sca1, c-kit, and Klf4 in the proximal descending aortas of western diet-fed *Apoe^−/−^* mice, with or without SB216763 treatment, as determined by real-time PCR. Wild-type (*Apoe^+/+^*) mice were used as controls (n = 6). (**e**) Total aortic calcium of the proximal descending aortas of *Apoe^−/−^* mice fed a western diet in combination with different doses of SB216763 (n = 6). (**f**) The association analysis of decreased expression of Osteopontin (OPN) with SB216763 treatment (n = 6) (**b**) was analyzed for statistical significance by an unpaired 2-tailed Student’s *t* test. (**c**,**e**) was analyzed for statistical significance by ANOVA with post hoc Tukey’s analysis. (**e**) was analyzed for statistical significance by the Pearson correlation coefficient. *, *p* < 0.05; **, *p* < 0.01; ***, *p* < 0.001.

**Figure 2 ijms-24-11638-f002:**
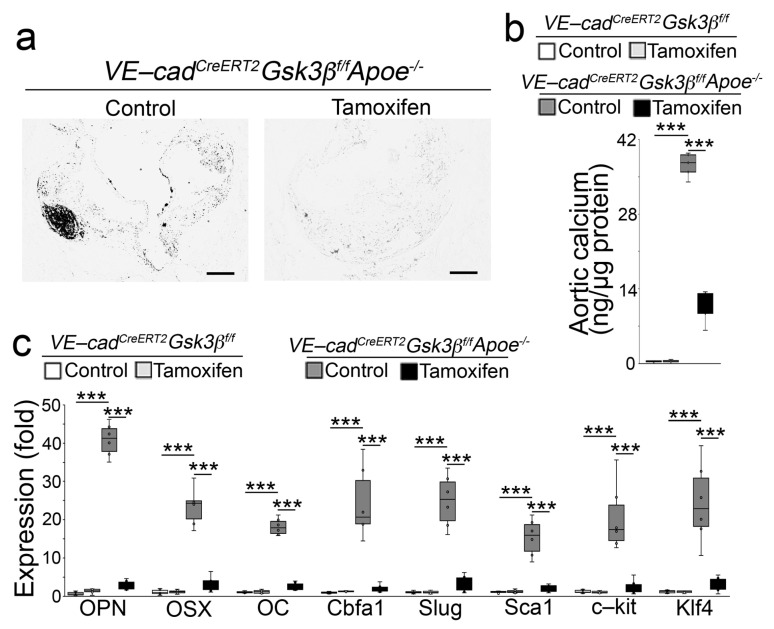
Deletion of *GSK3β* decreases atherosclerotic calcification in *Apoe^−/−^* mice. (**a**) Von Kossa staining of aortic sinus sections of the proximal ascending aortas of *VE-cadherin^CreERT2^GSK3β^flox/flox^Apoe^−/−^* mice fed a western diet with or without tamoxifen treatment. (**b**) Total calcium in the proximal descending aortas of *VE-cadherin^CreERT2^GSK3β^flox/flox^Apoe^−/−^* mice fed a western diet with or without tamoxifen treatment (n = 6). (**c**) The expression of the osteogenic markers Osteopontin (OPN), Osterix (OSX), Osteocalcin (OC), Cbfa1, and mesenchymal markers Slug, Sca1, c-kit, and Klf4 in the proximal descending aortas of *VE-cadherin^CreERT2^GSK3β^flox/flox^Apoe^−/−^* mice fed a western diet with or without tamoxifen treatment, as determined by real-time PCR (n = 6) (**b**,**c**) were analyzed for statistical significance by ANOVA with post hoc Tukey’s analysis. *** *p* < 0.001. Scale bar, 50 µm.

**Figure 3 ijms-24-11638-f003:**
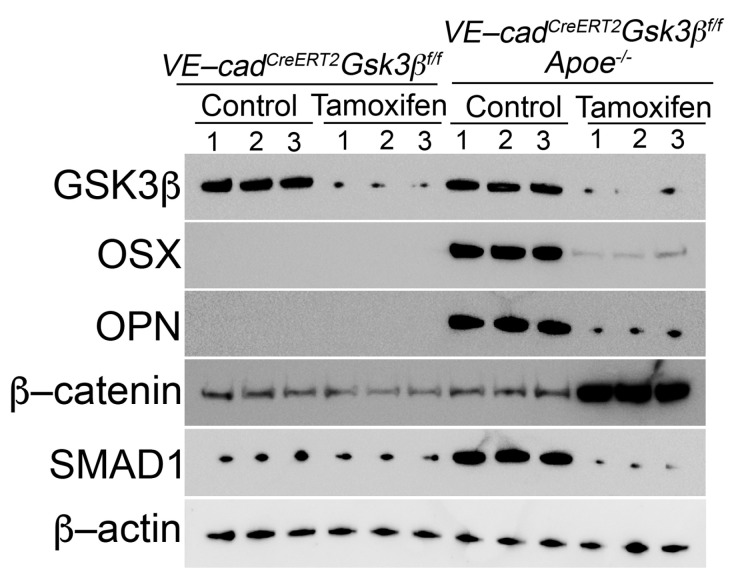
Immunoblotting of GSK3*β*, Osterix (OSX), Osteopontin (OPN), SMAD1 and *β*-catenin in the proximal descending aortas of *VE-cadherin^CreERT2^GSK3β^flox/flox/^* and *VE-cadherin^CreERT2^GSK3β^flox/flox^Apoe^−/−^* mice fed a western diet with or without tamoxifen treatment (n = 3). The numbers indicate samples from individual mice.

## Data Availability

No new data were created.

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
