# Peer review of "GSK3β Inhibition Ameliorates Atherosclerotic Calcification"

_ijms, 2023, doi:10.3390/ijms241411638_

Round 1

Reviewer 1 Report

In the present study, authors have determined that GSK3β inhibition or endothelial specific deletion of GSK3β reduces atherosclerotic calcification. My comments are us under:

1)      Please mention mice of which gender has been used in the current study. It is advisable to conduct the study in both the genders to avoid any bias.

2)      Authors have used 6 animals/ group. Please describe the rationale for choosing the number of animal/group. Any power analysis done?

3)      Though authors have checked changes in the Osteopontin and Osterix expression using RT-PCR but it’s worth exploring different markers for atherosclerotic calcification and EndMTs in the tissue sections using IHC or other technique.

Fine

Author Response

Reviewer 1

In the present study, authors have determined that GSK3β inhibition or endothelial specific deletion of GSK3β reduces atherosclerotic calcification. My comments are us under:

1)  “Please mention mice of which gender has been used in the current study. It is advisable to conduct the study in both the genders to avoid any bias.”

We added “Since similar atherosclerotic calcification occurred in both male and female Apoe−/− mice [7], we used mixed genders in each experimental group.” to the section of   Methods/Animals.

2) “Authors have used 6 animals/ group. Please describe the rationale for choosing the number of animal/group. Any power analysis done?”

We added “We used pwr R package to calculate the effect size and statistical power as we previously published [7]. The number of animals in each group was sufficient to reach more than 80% power and identify the difference in the formation of atherosclerotic calcification.” to the section of Methods/Animals.

3) “Though authors have checked changes in the Osteopontin and Osterix expression using RT-PCR but it’s worth exploring different markers for atherosclerotic calcification and EndMTs in the tissue sections using IHC or other technique.”

We added the aortic expression of osteogenic markers Osteocalcin, Cbfa1 and mesenchymal markers Slug, Sca1, c-kit and Klf4 to Figure 1 and 2. The results showed that GSK3 inhibition or endothelial-specific deletion of GSK3b reduced aortic expression of all these markers in Apoe-/- mice fed a western diet.

Reviewer 2 Report

Dear Editor,

In this manuscript, Cai X. et al. discussed the role of endothelial-mesenchymal transition (EndMT) in atherosclerotic calcification and the impact of glycogen synthase kinase-3 (GSK3) inhibition on this process. The main strength of this paper is based on how the author reported that GSK3 inhibition or endothelial-specific deletion of GSK3 reduces atherosclerotic calcification. Furthermore, the study reveals that the changes observed in β-catenin and SMAD1, as a result of GSK3 inhibition, in the aortas of Apoe-/- mice are similar to those surveyed in Mgp-/- mice. These findings suggest that GSK3 inhibition can effectively reduce vascular calcification in atherosclerosis, employing a similar mechanism as observed in Mgp-/- mice.

Altogether, this is a fascinating study, with well-designed experiments and conclusions. I recommend to accept this manuscript.

I found the quality of the English good. No significant issues were detected.

Author Response

Reviewer 2

“In this manuscript, Cai X. et al. discussed the role of endothelial-mesenchymal transition (EndMT) in atherosclerotic calcification and the impact of glycogen synthase kinase-3 (GSK3) inhibition on this process. The main strength of this paper is based on how the author reported that GSK3 inhibition or endothelial-specific deletion of GSK3 reduces atherosclerotic calcification. Furthermore, the study reveals that the changes observed in β-catenin and SMAD1, as a result of GSK3 inhibition, in the aortas of Apoe-/- mice are similar to those surveyed in Mgp-/- mice. These findings suggest that GSK3 inhibition can effectively reduce vascular calcification in atherosclerosis, employing a similar mechanism as observed in Mgp-/- mice. Altogether, this is a fascinating study, with well-designed experiments and conclusions. I recommend to accept this manuscript.”

     Thank you very much for the comments.
